# Vests with Radiative Cooling Materials to Improve Thermal Comfort of Outdoor Workers: An Experimental Study

**DOI:** 10.3390/nano14131119

**Published:** 2024-06-28

**Authors:** Yao Wang, Bohao Zhao, Hengxuan Zhu, Wei Yang, Tianpeng Li, Zhen Cao, Jin Wang

**Affiliations:** 1School of Energy and Environmental Engineering, Hebei University of Technology, Tianjin 300401, China; 202231304004@stu.hebut.edu.cn (Y.W.); 202331305013@stu.hebut.edu.cn (B.Z.); jin.wang@hebut.edu.cn (J.W.); 2School of Materials Science and Engineering, Hebei University of Technology, Tianjin 300401, China; yangwei@hebut.edu.cn; 3National Demonstration Center of Experimental Teaching for Ammunition Support and Safety Evaluation Education, Army Engineering University of PLA, Shijiazhuang 050000, China; ltp820325@163.com; 4Key Laboratory of PLA for Ammunition Support and Safety Evaluation, Army Engineering University of PLA, Shijiazhuang 050000, China; 5Department of Thermal and Fluid Engineering, University of Twente, 7522 NB Enschede, The Netherlands

**Keywords:** radiative cooling, thermal comfort, thermal sensation, wet bulb globe temperature, personal cooling vest, coating

## Abstract

This study focuses on improving human thermal comfort in a high-temperature outdoor environment using vests with a radiative cooling coating. The effects of coating thickness on the radiative cooling performance were first evaluated, and an optimal thickness of 160 μm was achieved. Then, six subjects were recruited to evaluate the thermal comfort in two scenarios: wearing the vest with radiative cooling coatings, and wearing the standard vest. Compared with the standard vest, the coated vest decreases the maximum temperature at the vest inner surface and the outer surface by 5.54 °C and 4.37 °C, respectively. The results show that thermal comfort is improved by wearing radiative cooling vests. With an increase of wet bulb globe temperature (WBGT), the improving effects tend to decline. A significant improvement in human thermal comfort is observed at a WBGT of 26 °C. Specifically, the percentage of thermal sensation vote (TSV) wearing the cooling vest in the range of 0 to 1 increases from 29.2% to 66.7% compared with that of the untreated vest. At the same time, the average value of thermal comfort vote (TCV) increases from −0.5 to 0.2.

## 1. Introduction

Thermal comfort is a subjective sensation of individuals related to the surrounding environment. Thermal comfort mainly depends on the external environmental parameters under a certain amount of human activity and clothing [1]. The increment in average seasonal temperatures has a negative impact on thermal comfort for outdoor individuals, such as field personnel, firefighters, and construction workers. Outdoor laborers frequently suffer from heat stress reactions, which lead to low work efficiency and health problems [2,3,4]. Therefore, it is urgent to address the harm caused by prolonged physical labor in a high-temperature environment.

Two cooling approaches are widely used to maintain thermal comfort. One is active cooling, which achieves an excellent cooling effect at the cost of energy consumption [5]. Specifically, ventilation and air conditioning systems used for active space cooling typically have high capital and operating costs, which are impractical for outdoor services [6]. Liquid cooling [7,8], air cooling [9,10,11], semiconductor cooling [12,13], and phase change material cooling [14] textiles and wearables used for personal active cooling consume less energy. Still, current wearables are bulky and require regular replacement of accessories, making them unsuitable for outdoor workers to wear for a long period. The other is passive daytime radiative cooling (PDRC), which is very promising for personal thermal management because of its excellent cooling effect [15]. It radiates heat to outer space and reflects the sunlight without extra power to achieve cooling [16,17,18,19,20,21,22]. At present, various intelligent fabrics enabled by nanofiber membranes, microfiber fabric, and coated fabric have been successfully utilized in personal thermal management. Zhang et al. [23] developed a passive cooling layered metamaterial fabric based on a nanofiber membrane for moisture absorption and perspiration. The metamaterial fabric showed high solar reflectance emissivity and remarkable moisture permeability. The human body temperature covered by metamaterial cooling fabric developed by Zeng et al. [24] was about 4.8 °C lower than that covered by cotton fabric. Cheng et al. [25] prepared a radiative cooling coating with the biomimetic structure of human natural skin wrinkles, achieving a sub-ambient cooling temperature of 8.1 °C at the most. Compared to other nano-based methods, nano-coating for surface modification offers strong operability at a low cost while achieving excellent cooling effects. Figure 1 shows the working schematic for radiative cooling coated fabrics. When sunlight strikes the coated fabric, it cools itself by reflecting the sunlight and emitting heat energy to cold outer space through the atmospheric window. Zhang et al. [26] employed a simple and scalable spraying strategy to fabricate colorful radiative cooling cotton fabrics, which achieved a temperature difference of 5.6 °C compared to bare cotton fabrics. Xiao et al. [27] enhanced the cooling effects of safety helmets with a radiative cooling coating. Under direct sunlight, the internal temperature of the improved helmet is lower than that of the ordinary white helmet, and the maximum temperature difference is 12.2 °C. Zhang et al. [28] experimentally compared the thermal comfort of two reduced-size tent models. The results indicated that the average temperature in the tent model using reflective materials was 4.8 °C lower than that of a regular tent. In addition, the PDRC materials attached to the fabric, and the coating thickness are crucial factors in determining the cooling effect of coated fabrics [19]. Zhong et al. [29] fabricated a multifunctional cotton fabric for radiative outdoor cooling using spectrally selective aluminum phosphate particles. Since the thicker coating results in uneven pore sizes and stiffening of the textile, Cui et al. [30] selected the ideal thickness when considering the preparation process and the flexibility of the coated textile.

Considering the impact of solar radiation in outdoor high-temperature conditions, WBGT is widely used to evaluate the thermal load of outdoor working environments. The higher WBGT means a stronger thermal load in the workplace. The evaluation of human thermal comfort is usually conducted through subjective questionnaire surveys. Mahgoub et al. [31] evaluated the thermal comfort of the outdoor environment by adopting WBGT as a thermal stress index. Yang et al. [32] studied the effect of local cooling on human thermal comfort for different parts of the torso under high-temperature environments. According to a human thermal sensation questionnaire survey, upper back cooling significantly improved the thermal sensation and comfort of subjects in hot environments. Ghani et al. [33] studied the applicability of seven different thermal comfort indicators in evaluating outdoor thermal comfort, and it was found that the WBGT index was consistent with the actual TSV data.

Although PDRC materials have been developed and their cooling properties tested, few studies have examined the integration of PDRC materials with basic clothing to assess human thermal comfort. Inspired by the above research, this work developed a novel radiative cooling vest with the coating strategy, and the thermal load of the outdoor environment was evaluated with the WBGT index. Through an experiment of radiative cooling performance in a high-temperature outdoor environment, the optimal thickness of the coating was determined to modify the surface property of the vest. The cooling performance of the vest with radiative cooling was experimentally investigated compared with the standard vest. In addition, the TSV and TCV were collected and analyzed to evaluate the human thermal comfort after the outdoor human experiment at different WBGTs.

## 2. Materials and Methods

This section introduces the experimental materials and environments, the cooling performance test system, the experimental method, and the characterization of coatings.

### 2.1. Experimental Materials and Environments

The nano-coating used in the experiment was commercially purchased from Shanghai Jingfu building Materials Co., Ltd. (Shanghai, China). The main components of the nano-coating materials are silica and water-based acrylic acid. SiO_2_ with Si−O−Si structure exhibits a phonon–polaron resonance phenomenon in the atmospheric window, which enhances its emissivity in the mid-infrared band. In addition, SiO_2_ possesses a high reflectivity in the solar spectral range due to its strong particle scattering effect. The coating radiates heat to space through the atmospheric window to achieve daytime radiative cooling. At the same time, the material receives less solar radiation due to high reflectivity. The combined effect of the two aspects achieves the cooling effect. The standard vest is made of polyester and is covered with soft inserts of ethylene vinyl acetate copolymer. It also features adjustable shoulder straps.

The experiment was conducted from mid-August to early September in 2023 to ensure that the outdoor environmental conditions met the requirement of the high-temperature experiment. The experimental site was open and unobstructed to avoid the influence of surrounding objects on the experimental conditions. The performance test of radiative cooling was performed at 10:00–16:00 on the side of a college building at Hebei University of Technology in Tianjin, China (39.1° N, 117.2° E). The PDRC measurement of coated fabrics was conducted under direct sunlight with a solar intensity of 910.6 W/m^2^, an ambient temperature of 39 °C, a relative humidity of 42%, a wind speed of 2.2 m/s, and an atmospheric pressure of 1.001 × 10^5^ Pa. The PDRC measurement of the coated vest was conducted under direct sunlight with a solar intensity of 738.2 W/m^2^, an ambient temperature of 39 °C, a relative humidity of 32%, a wind speed of 1.6 m/s, and an atmospheric pressure of 1.004 × 10^5^ Pa. The human comfort experiment was carried out on a road in Hebei University of Technology, and the experiment period was from 11:00 to 15:00. During the experiment, the cloud cover ranged from 0–30% in the sky.

### 2.2. Experimental Methods

This section introduces the three experimental groups of this work, including radiative cooling experiments with coatings of different thicknesses, comparison experiments on the cooling performance of two vests, and thermal comfort experiments.

#### 2.2.1. The Radiative Cooling Experiments with Coatings of Different Thicknesses

The fabric samples were coated with different coating thicknesses of about 100 µm, 160 µm, 220 µm, and 300 µm, respectively. The fabric sample size was 6 cm×6 cm. The outdoor test platform for the radiative cooling performance of coated fabrics consists of a test box for shielding forced convection heat transfer, a data acquisition system, thermocouples, and polyethylene film. The coated fabrics and the bare fabric were placed in the shielding box. The inter-surface temperature of fabrics was measured in real-time. Meanwhile, the meteorological parameters were collected and recorded by the outdoor weather station, including the solar irradiance, ambient temperature, wind speed, and air humidity in the experimental environment. When fabrics were exposed to direct sunlight, a low-density polyethylene film (LDPE) was placed on the top of the hollow polystyrene foam box to minimize the impact of heat conduction and convection between the fabric and the environment on the accuracy of the experimental data. In order to prevent the box from absorbing solar radiation and causing an increment of internal temperature, the surface of the test box was covered with Al foil with high reflectivity. The schematic diagram of the test device and the photograph of physical objects are shown in Figure 2.

#### 2.2.2. The Comparison Experiments on the Cooling Performance of Two Vests

The standard vest was coated with the radiative cooling coating of the correct thickness and dried under natural conditions, obtaining a vest with radiative cooling performance. The cooling vest covers the torso part of the body, including the chest, abdomen, back, and waist areas. Table 1 shows the differences between the coated and standard vests. Figure 3a shows two vests worn on manikins for radiative cooling performance testing. Eight temperature measurement points were pasted on the surface of each vest. The inner surface temperature of the vest was measured by 1–7 temperature measurement points, and the eighth temperature measurement point was placed on the outer surface of the vest. The measurement points of the vests and a schematic diagram of an experiment are shown in Figure 3b. The temperature of the vest and meteorological parameters were measured and recorded in real-time under continuous outdoor experiments for 6 h.

#### 2.2.3. The Thermal Comfort Experiments

The experiments involved four distinct WBGTs: 26, 28, 30, and 32 °C, which represent potential thermal conditions in a natural outdoor working environment in summer. Radiative cooling vests and standard vests were used for the outdoor experiment on thermal comfort.

Before the experiments, the minimum number of participants was estimated to be six (total sample size/number of measurements) using the G*Power software 3.1.9.2. The effect size, α and statistical power (1−β) are 0.4, 0.05, and 0.8, respectively (α and β are the probability of type I and II errors, respectively). The subjects selected for the experiment were all healthy men of around 23 years old. Six healthy college students (age: 23 ± 1 year, height: 176 ± 3 cm, weight: 69 ± 7 kg, BMI: 21.0 ± 2.7 kg/m^2^) were involved in the subjective questionnaires on thermal comfort at four different WBGTs. The participants were informed not to take any alcohol, caffeine, or medical drugs for at least 24 h before the experiments. The subjects wore the same basic outfit under a vest, including a cotton T-shirt, knee-high shorts, socks, underwear, and thin sneakers. The clothing insulation was approximately 0.5 clo [34]. The study was approved by the University Ethics Committee and all participants provided written informed consent before the experiments. The guidelines of the Declaration of Helsinki were followed in the experiment [35]. The personal information of the subjects is shown in Table 2.

The experiment aimed to investigate the effect of radiative cooling vests on human comfort in high-temperature environments. Considering this, the seven-point scales were applied to the evaluation of thermal sensation with reference to ISO 10551 [34], in which the vote ranges from −3 (cold) to 3 (hot). The five-point scales ranging from −2 (very uncomfortable) to 2 (very comfortable) were applied to the evaluation of thermal comfort [36]. Thermal evaluation scales are shown in Figure 4a. The content of the questionnaires and the meaning of the scales were explained in detail before the experiments. Precise responses were made during experiments to ensure that the participants fully understand the meaning. The subjects were asked to retain one decimal point when filling out the questionnaires. The WBGT was measured and recorded continuously. Figure 4b shows the experimental scenario of the thermal comfort experiment.

The comfort experiment consists of four stages: In the first stage, the participants are dressed in the same basic outfits without the vest. Participants sit quietly for 20 min to adapt to the thermal environment of experiments. In the second stage, participants walk at 4 km/h for 25 min with the standard vest. In the third stage, the participants take off the standard vests and rest in a shaded room for 20 min. In the fourth stage, the participants put on the radiative cooling vests and keep walking at a speed of 4 km/h for 25 min. The entire experiment lasts 100 min. Each vest is worn for 25 min, and participants are asked to fill out a questionnaire about their subjective heat evaluation in the last five minutes. During the experiment, subjects walk at a speed of 4 km/h in order to simulate the moderate labor intensity of workers. The experimental procedure is shown in Figure 5.

SPSS statistics software (version 26) was used in the statistical analysis. In the experiment, one-way analysis of variance was used for comparing the thermal comfort of subjects wearing standard and refrigerated vests under each specific wet bulb globe temperature. The samples of one-way analysis satisfied normality, independence, and homogeneity of variance. Results were considered statistically significant at *p* < 0.05. To assess the statistical significance of differences in thermal comfort between standard vests and cooling vests, a paired-sample t-test was performed to analyze the significant difference in relevant variables under the standard vest and cooling vest conditions. The samples of the paired t-test satisfied the pairing condition, had normally distributed differences, exhibited independence between pairs, and assumed equal variance of the differences. The symbol “*” indicates a significant difference (*p* < 0.05), and the symbol “**” indicates a highly significant difference (*p* < 0.01) in the figures.

The fabric samples to be tested were cut into a size of 2 cm × 2 cm, and the UV–Visible–Near-infrared photometer (VARIAN Cary5000, Palo Alto, CA, USA) with integrating sphere attachment was used to test the reflectivity of the samples in the wavelength range of 0.3–25 µm. The emissivity was measured by a Fourier infrared spectrometer (Model Bruker 5700, Ettlingen, Germany) equipped with a gold integrating sphere. The test wavelength range was set to be 2.5–16.0 µm. The experimental data were collected by an Agilent 34970A data acquisition instrument (Santa Clara, CA, USA). In the experiment, an Omega thermistor was connected to the module slot of the device to measure the temperature. The temperature of the fabric was measured by K-type thermocouples (OMEGA, Stamford, CT, USA) with an accuracy of ±0.2 °C, a use range of −50–200 °C, and a relative error range of 0.2–1%. The temperature and humidity of the ambient air were recorded and measured with a temperature and humidity sensor (JXBS-3001-TH-5-U, JXCT Electronics Co., Ltd., Weihai, China), at a position of 1.5 m above the ground. Their specific technical parameters were introduced as follows: a temperature accuracy of ±0.2 °C, a humidity accuracy of ±2%, a temperature range of −20–60 °C, and a humidity range of 5–95%, a temperature relative error range of 0.2–2%, and a humidity relative error range of 0.2–2%. The wind speed was measured with a thermal anemometer (KA23, Kano Instrument (Shanghai) Co., Ltd., Shanghai, China). The measuring height was 1.5 m above the ground, and the measuring wind speed ranged from 0 to 50 m/s with an accuracy of ±2% and a relative error range of 0.1–2%. WBGT in the experimental environment was measured with a bulb globe thermometer (AZ8778, Zhongshan Runjing Technology Co., Ltd., Zhongshan, China). The measuring height was 1.7 m away from the roof, and the diameter of the black ball was 75 mm. The measurement accuracy was ±1.5 °C, the range of the bulb globe temperature was 0–80 °C, and the relative error ranged from 0.5% to 2.5%. The solar radiation intensity of the experimental environment was measured by a solar power meter (TES1333, Taiwan Taishi Electronic Industry Co., Ltd., Taiwan, China) with a measurement resolution of 0.1 W/m^2^ and an error value of ±10 W/m^2^ and a relative error range of 0.1–5%.

## 3. Results

In this section, the optical properties of radiative cooling material are analyzed and the radiative cooling performance of the coating revealed. Human thermal comfort was evaluated at different WBGT levels.

### 3.1. Optical Properties of Coated Fabric

The reflectivity of the prepared coating in the solar spectral region is a crucial factor affecting the performance of daytime radiative cooling [15]. The reflectivities of five fabrics with different coating thicknesses are shown in Figure 6. In the visible light region (VI, 0.38–0.78 µm), the reflectivity of the coated fabrics is apparently higher than the bare fabric. In the near-infrared region (NIR, 0.78~2.5 µm), the fabric with 160 µm-thick coating possesses the highest reflectivity of 82.53%.

### 3.2. Effect of Coating Thicknesses on Radiative Cooling Performance

An outdoor experiment was conducted to compare the cooling performance of the fabrics with coatings of different thicknesses. The measured meteorological parameters, the inner surface temperature of the fabric, and the air temperature inside the test box are shown in Figure 7a. The fabrics with coating thicknesses of 160 µm and 220 µm have lower temperatures than those of the other coated fabrics for most of the time. The fabric temperature was reduced by 12.22 °C maximum with the coating of 160 µm in thickness, compared with the air temperature inside the test box. The average temperatures of the fabrics with different coating thicknesses are shown in Figure 7b. It can be seen that the coating with a thickness of 160 um has a maximum average temperature 5.87 °C lower than the air temperature inside the test box. It can be concluded that the fabric with a coating thickness of 160 µm possesses the optimal radiative cooling performance. By the time the fabrics were exposed to the sun for 170 min, the temperatures of the inner surface of the fabrics decreased significantly due to the reduction of solar radiation intensity (see the red circle in Figure 7a). The decrease in solar radiation intensity is caused by a sudden increase in cloud cover in the sky. In addition, the fabric with a coating thickness of 160 µm is softer than the fabric with a coating thickness of 220 µm and 300 µm, and the coated fabric surface possesses higher smoothness and better uniform continuity. From the above experimental results, it can be seen that the fabric with a coating thickness of 160 µm shows a higher cooling effect. At the same time, good economics for preparing the coated fabric are clearly reflected compared with thicker coating fabrics.

### 3.3. Radiative Cooling Performance of Vests

Based on the results of the above studies, the coating thickness of 160 µm was selected for the radiative cooling vest. Figure 8 shows the temperature variation of two vests when manikins are placed in the same outdoor environment. The temperatures at the inner and outer surfaces of the radiative cooling vest are lower than those of the standard vest. The maximum temperature difference recorded on the inner surfaces of the two vests was 5.54 °C, while the maximum temperature difference on their outer surfaces was 4.37 °C.

### 3.4. Thermal Evaluation Analysis

Statistical analysis was conducted on the thermal evaluation results of subjects wearing two vests in outdoor environments when the WBGTs were 26 °C, 28 °C, 30 °C, and 32 °C. The standard vest is named SV, and the radiative cooling vest is named CV, as shown in the following figures. The results of *p* < 0.01 is marked with a symbol “**”, and *p* < 0.05 is marked with a symbol “*”. Figure 9 shows the results of TSV and TCV at different WBGTs. Figure 9a illustrates that the overall TSVs continue to rise as WBGT increases for subjects wearing the two vests. By applying the radiative cooling coating to the vest, the average TSVs decrease from approximately 1.5 to 0.8 at WBGT of 26 °C, from about 1.7 to 1.1 at WBGT of 28 °C, and from around 1.9 to 1.6 at WBGTs of 30 °C. The dispersions of TSV are relatively scattered for the WBGTs of 26 °C, 28 °C, and 30 °C. The coated vest improves the comfort of people in a hot environment. The distribution of TSV is relatively concentrated at WBGT of 32 °C, and the mean value of TSV is above 2 for the two vests. This result indicates that the radiative cooling vest has little effect on improving human thermal sensation at high WBGT. Figure 9b shows that overall the TCVs continue to reduce as the WBGT increases. The mean TCV varies from −0.5 to 0.2 by adding a radiative cooling coating on the standard vest at WBGT of 26 °C with a significantly improved effect on thermal comfort. The mean TCV in the radiative cooling vest increases by 0.4 compared with the standard vest at WBGT of 28 °C. The mean TCVs are below −1 for the radiative cooling vests at WBGTs of 30 °C and 32 °C, which indicates that the radiative cooling coating has no significant effect on the improvement of human thermal comfort at high WBGT. In addition, the decrements in TSV and TCV become less between the SV and CV conditions with the increase of WBGT. This result confirms that the continuous exposure to a high WBGT environment weakens the cooling effect of radiative cooling vests.

The percentage distribution of thermal evaluation votes at different WBGTs is shown in Figure 10. TSV ≥ 2 or TSV ≤ −2 is an unacceptable range for human thermal sensation, while −1 ≤ TSV ≤ 1 is an acceptable range. Figure 10a shows that the thermal sensation of wearing a radiative cooling vest is lower than that of wearing a standard vest at all WBGTs. When WBGT is 26 °C, the radiative cooling vest significantly improves thermal comfort with an increase of percentage from 29.2% to 66.7% in the range of 0–1. The proportion of unacceptable TSV to the human body decreases from 41.7% to 16.7% at the WBGT of 28 °C. When WBGT is 30 °C, the ability of radiative cooling vests to improve thermal comfort declines with an increase from 50% to 58.4% in the range of 1–2. The thermal sensations of wearing the two vests are nearly identical at a WBGT of 32 °C. The thermal comfort results are shown in Figure 10b. TCV in the range of 0–2 indicates comfort, and in the range of −2–0 indicates discomfort. Similar to thermal sensation results, as the WBGT increases, the thermal comfort becomes increasingly worse for subjects wearing two vests. When WBGT is 26 °C, the proportion of votes in which the human body feels comfortable increases from 25% to 75% with the radiative cooling vest, which significantly improves human thermal comfort. However, under the latter three WBGTs, the cooling effect of the radiative cooling vest is weakened gradually. Specifically, the percentage of comfort ranges increases from 4.2% to 20.8% at a WBGT of 28 °C. When WBGTs are 30 °C and 32 °C, the percentages of uncomfortable votes are all at 100% for subjects wearing two vests.

## 4. Conclusions

In this study a radiative cooling vest was developed based on a working suit, which was easily made at low cost. The human thermal comfort was evaluated in conditions of wearing radiative cooling vests and standard vests. The radiative cooling performance was tested for coated fabrics with different thicknesses. Radiative cooling vests and standard vests were compared, concerning the inner and outer surface temperatures in an outdoor high-temperature environment. The radiative cooling vest significantly improved human thermal comfort. The present study leads to the following conclusions:

(1) The fabric with a thickness of 160 µm has a remarkable radiative cooling effect with an average temperature of 5.87 °C lower than the ambient temperature.

(2) Compared with the standard vest, the maximum temperatures at the inner and outer surfaces of the radiative cooling vest are reduced by 5.54 °C and 4.37 °C, respectively.

(3) Thermal comfort of the human body is improved by wearing the radiative cooling vests at different wet bulb globe temperatures, but the degree of improved cooling decreases with the increase of wet bulb globe temperature.

(4) A significant improvement of human thermal comfort was achieved at a wet bulb globe temperature of 26 °C. Compared with the standard vest, the percentage of thermal sensation vote in the range of 0–1 increased from 29.2% to 66.7%, and the mean thermal comfort vote increased from about −0.5 to 0.2 by applying the radiative cooling coating to the vest.

The cooling vest developed in this work still suffers from poor air permeability, and the cooling effect requires further enhancement. To address these issues, it may be necessary to develop multiple types of cooling clothing, such as by adding fans or developing new cooling materials. However, the significant improvement in comfort provided by the cooling vest is sufficient for its application for departments whose work activities are carried out outdoors, such as construction workers, traffic police, cleaners, and other outdoor workers. The purpose of removing heat from the human body and improving human performance can be achieved by wearing a cooling vest representing wearable technology.

## Figures and Tables

**Figure 1 nanomaterials-14-01119-f001:**
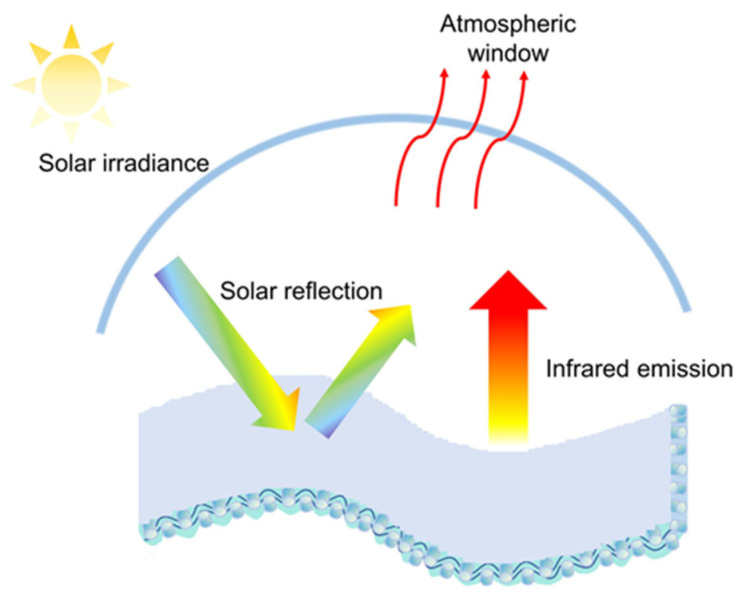
The working schematic of a radiative cooling coated fabric.

**Figure 2 nanomaterials-14-01119-f002:**
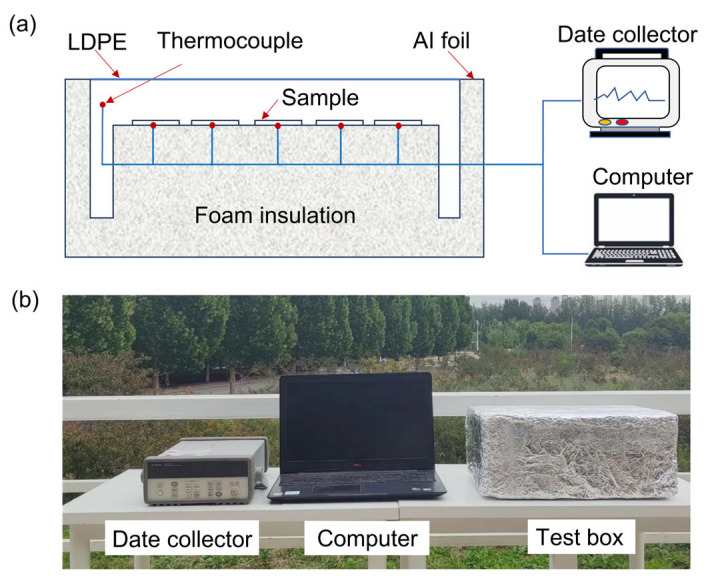
Thermal measurement of the coated fabrics: (**a**) the schematic of the experimental device, (**b**) the photograph of the experimental device.

**Figure 3 nanomaterials-14-01119-f003:**
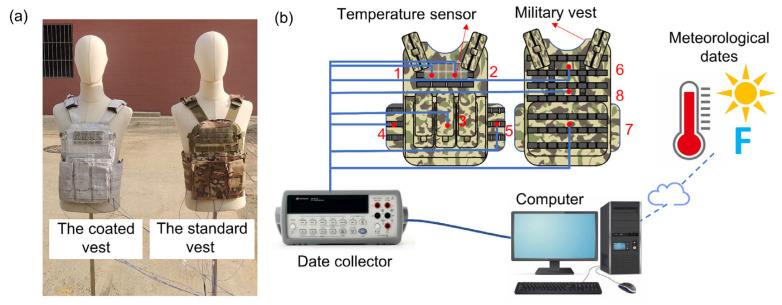
The experiment on cooling performance of vests: (**a**) the photograph of two vests, (**b**) the schematic diagram of the experimental system (1–8 are the location of thermocouples).

**Figure 4 nanomaterials-14-01119-f004:**
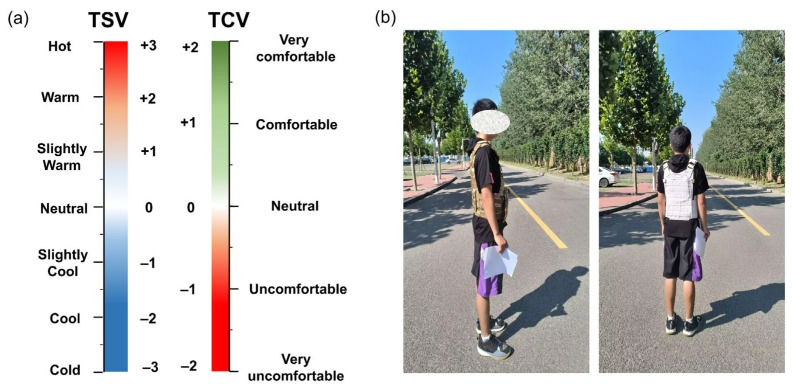
The experiment on human comfort: (**a**) thermal evaluation scales, (**b**) the experimental scenario.

**Figure 5 nanomaterials-14-01119-f005:**
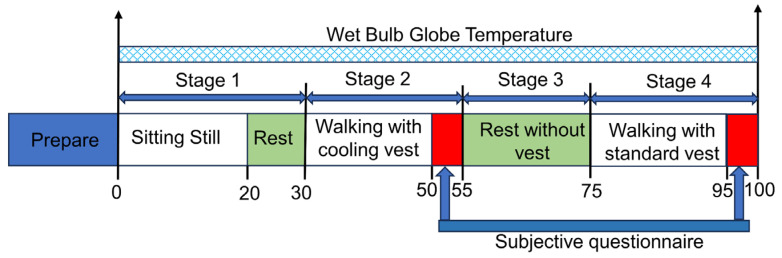
The procedure of experiments.

**Figure 6 nanomaterials-14-01119-f006:**
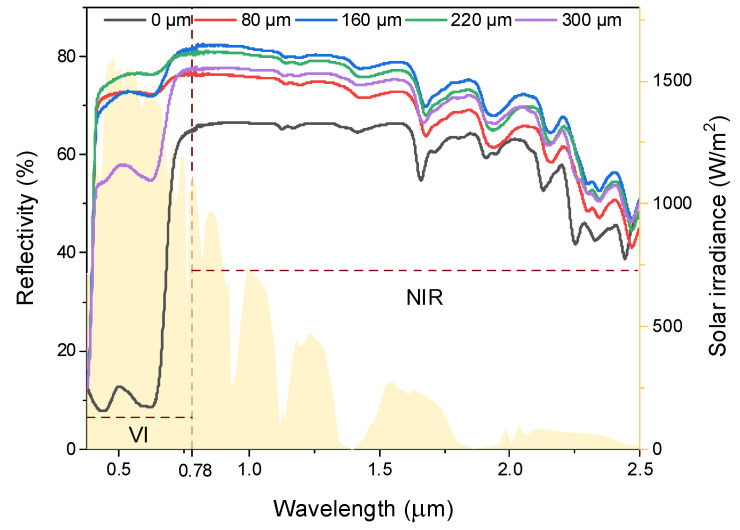
Reflectance spectra of fabrics with different coating thicknesses.

**Figure 7 nanomaterials-14-01119-f007:**
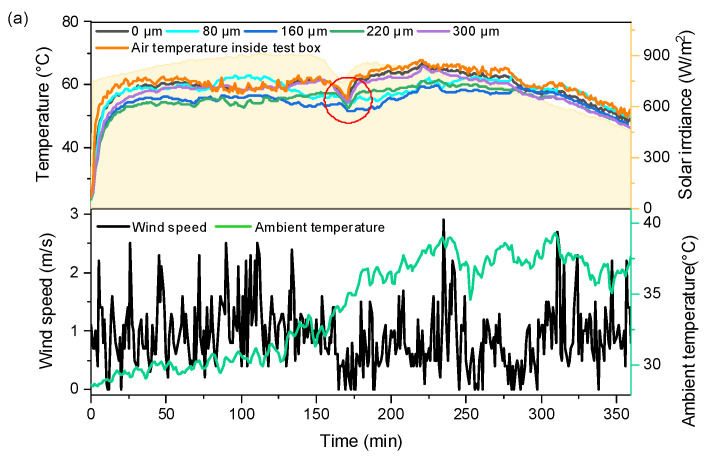
Radiative cooling performance of fabrics with different coating thicknesses: (**a**) internal surface temperatures of fabrics and weather conditions, (**b**) average temperatures of fabrics with different coating thicknesses.

**Figure 8 nanomaterials-14-01119-f008:**
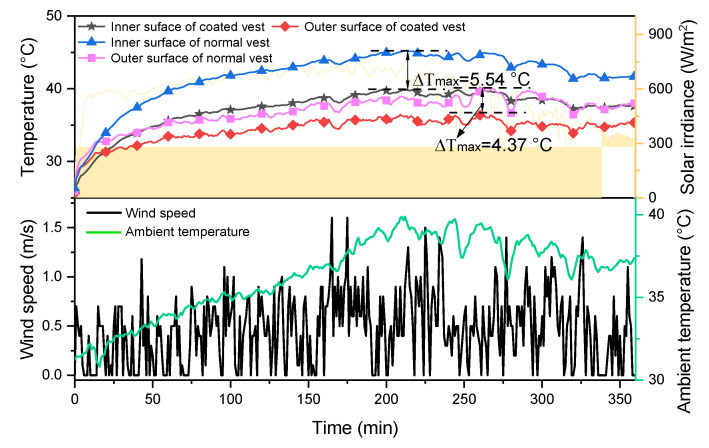
Comparison of radiative cooling performance of two vests.

**Figure 9 nanomaterials-14-01119-f009:**
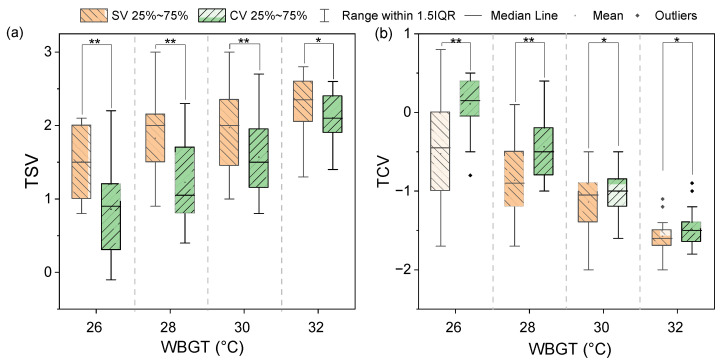
Thermal evaluation at different WBGTs: (**a**) TSV, (**b**) TCV.

**Figure 10 nanomaterials-14-01119-f010:**
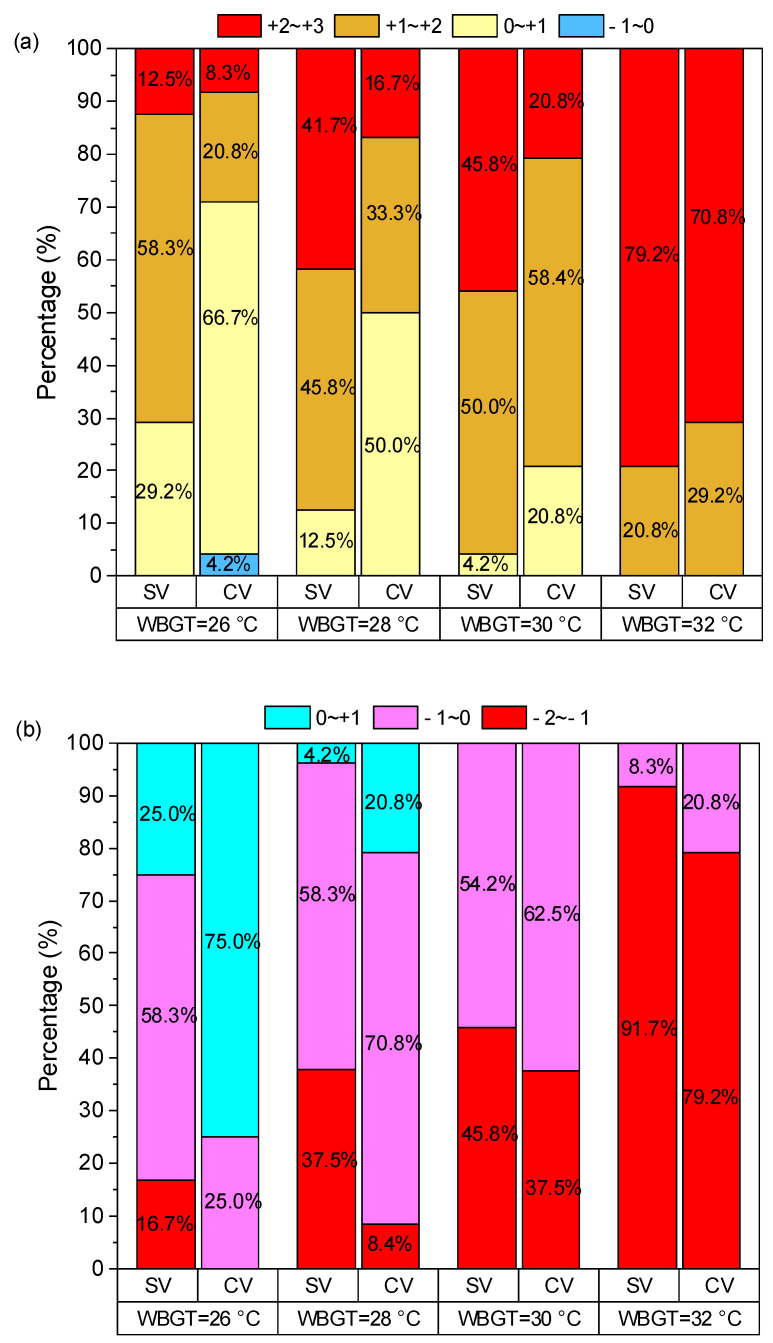
Percentage distribution of thermal evaluation votes at different WBGTs: (**a**) TSV, (**b**) TCV.

**Table 1 nanomaterials-14-01119-t001:** The comparison for the coated and standard vests.

Type	Coating Thickness (µm)	Atmospheric Window Emissivity (%)	Reflectivity (%)	Absorptivity (%)
Coated vests	160	90.2	82.5	17.5
Standard vests	0	89.7	65.1	34.9

**Table 2 nanomaterials-14-01119-t002:** The personal information of subjects.

Subjects	Age (years old)	Height (cm)	Weight (kg)	BMI (kg/m^2^)
A	22	174	62	20.5
B	22	173	68	22.7
C	24	175	70	22.8
D	23	176	69	22.3
E	23	179	67	20.9
F	24	178	75	23.7

## Data Availability

Data are available in a publicly accessible repository.

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
