# Peer review of "Vests with Radiative Cooling Materials to Improve Thermal Comfort of Outdoor Workers: An Experimental Study"

_nanomaterials, 2024, doi:10.3390/nano14131119_

Round 1

Reviewer 1 Report

Comments and Suggestions for Authors

1.The thermal conditions of the experiment must be specified in full (temperature, atmospheric pressure, air velocity, air humidity, solar activity, atmospheric pressure).

2.The material and design of the vests should be described as accurately as possible. From the composition of materials and coatings to the parameters of changing each section and parts of the product. The overall thermal effect of clothing depends on all conditions at the same time.

3.The authors should provide a calculation for the number of testers taking into account the estimation of the error of the results (statistical parameters of experiment planning).

4. The parameters of the measuring equipment should be presented in detail, indicating the calculated coefficients for estimating the actual measurement errors.

Author Response

Comment 1: The thermal conditions of the experiment must be specified in full (temperature, atmospheric pressure, air velocity, air humidity, solar activity, atmospheric pressure). 

Reply 1:Thank you for this comment. We have revised the manuscript as follows:“The experiment is conducted from mid-August to early September in 2023 to ensure that the outdoor environmental conditions meet the requirement of the high-temperature experiment. The experimental site is open and unobstructed to avoid the influence of surrounding objects on the experimental conditions. The performance test of radiative cooling is performed at 10:00–16:00 on the side of a college building at Hebei University of Technology in Tianjin, China (39.1 N, 117.2 E). The PDRC measurement of coated fabrics is conducted under direct sunlight with a solar intensity of 910.6 W/m2, an ambient temperature of 39 °C, a relative humidity of 42%, a wind speed of 2.2 m/s, and an atmospheric pressure of 1.001×105 Pa. The PDRC measurement of the coated vest is conducted under direct sunlight with a solar intensity of 738.2 W/m2, an ambient temperature of 39 °C, a relative humidity of 32%, a wind speed of 1.6 m/s, and an atmospheric pressure of 1.004×105 Pa. The human comfort experiment is carried out on a road in Hebei University of Technology, and the experiment period is from 11:00 to 15:00. During the experiment, the cloud cover ranges from 0-30% in the sky.”

Comment 2: The material and design of the vests should be described as accurately as possible. From the composition of materials and coatings to the parameters of changing each section and parts of the product. The overall thermal effect of clothing depends on all conditions at the same time.

Reply 2: Thanks for your valuable suggestion. The material and design of the vests in this work have been added to the manuscript in detail as follows: “The nano-coating used in the experiment is commercially purchased. The main components of the nano-coating materials are silica and water-based acrylic acid. SiO2 with Si−O−Si structure exhibits a phonon-polaron resonance phenomenon in the atmospheric window, which enhances its emissivity in the mid-infrared band. In addition, SiO2 possesses a high reflectivity in the solar spectral range due to its strong particle scattering effect. The coating radiates heat to space through the atmospheric window to achieve daytime radiative cooling. At the same time, the material receives less solar radiation due to high reflectivity. The combined effect of the two aspects achieves the cooling effect. The standard vest is made of polyester and is covered with soft inserts of ethylene vinyl acetate copolymer. It also features adjustable shoulder straps.

Table 1. The comparison between the coated and standard vests.

Type Coating thickness (μm) Atmospheric window emissivity (%) Reflectivity (%) Absorptivity (%)
Coated vest 160 90.2 82.5 17.5
Standard vest 0 89.7 65.1 34.9

Comment 3: The authors should provide a calculation for the number of testers taking into account the estimation of the error of the results (statistical parameters of experiment planning).

Reply 3: Thank you for your valuable suggestion. We have provided a calculation for the number of testers. “Before the experiments, the minimum number of participants was estimated to be six (total sample size/number of measurements) using the G*Power software. The effect size, α and statistical power (1-β) are 0.4, 0.05, and 0.8 respectively (α and β are the probability of type I and II errors, respectively). The subjects selected for the experiment are all healthy men of around 23 years old. Six healthy college students (age: 23±1 years old, height: 176±3 cm, weight: 69±7 kg, BMI: 21.0±2.7 kg/m2) are involved in the subjective questionnaires on thermal comfort at four different WBGTs. The participants are informed not to take any alcohol, caffeine, or medical drugs for at least 24 hours before the experiments. The subjects wear the same basic outfit under a vest, including a cotton T-shirt, knee-high shorts, socks, underwear, and thin sneakers. The clothing insulation is approximately 0.5 clo [34]. The study is approved by the University Ethics Committee and all participants provide written informed consent before the experiments. The guidelines of the Declaration of Helsinki are followed in the experiment [35]. The personal information of the subjects is shown in Table 2.” 

Comment 4: The parameters of the measuring equipment should be presented in detail, indicating the calculated coefficients for estimating the actual measurement errors.

Reply 4: According to your suggestion, we have added a detailed description of the parameters concerning the measuring equipment, and relative errors as the calculated coefficients have been provided for estimating the actual measurement errors.

“The fabric samples to be tested are cut into a size of 2 cm×2 cm, and the UV-visive-near-infrared photometer (VARIAN Cary5000, made in the USA) with integrating sphere attachment is used to test the reflectivity of the samples in the wavelength range of 0.3~25 µm. The emissivity is measured by a Fourier infrared spectrometer (Model Bruker 5700, made in Germany) equipped with a gold integrating sphere. The test wavelength range is set to be 2.5~16.0 µm. The experimental data are collected by an Agilent 34970A data acquisition instrument. In this experiment, an Omega thermistor is connected to the module slot of the device to measure the temperature. The temperature of the fabric is measured by K- type thermocouples (OMEGA) with an accuracy of±0.2 °C, a use range of -50 °C ~200 °C, and a relative error range of 0.2%~1%. The temperature and humidity of the ambient air are recorded and measured with a temperature and humidity sensor (JXBS-3001-TH-5-U, JXCT), at a position of 1.5 m high from the ground. Their specific technical parameters are introduced as follows: a temperature accuracy of ±0.2 °C, a humidity accuracy of ±2%, a temperature range of -20~60 °C, a humidity range of 5~95%, a temperature relative error range of 0.2%~2%, and a humidity relative error range of 0.2%~2%. The wind speed is measured with a thermal anemometer (KA23, KANOMAX). The measuring height is 1.5 m from the ground, the measuring wind speed ranges from 0 to 50 m/s with an accuracy of ±2% and a relative error range of 0.1%~2%. WBGT in the experimental environment is measured with a black ball thermometer (AZ8778). The measuring height is 1.7 m away from the roof, and the diameter of the black ball is 75 mm. The measurement accuracy is ±1.5 °C, the range of the black ball temperature is 0-80 °C and the relative error ranges from 0.5% to 2.5%. The solar radiation intensity of the experimental environment is measured by a solar power meter (TES1333) with a measurement resolution of 0.1 W/m2 , an error value of ±10 W/m2 and a relative error range of 0.1%~5%.”

Reviewer 2 Report

Comments and Suggestions for Authors

The paper is interesting but the introduction provides a comprehensive background. However, the authors have to Include a brief discussion on the limitations of previous studies on PDRC to better highlight the novelty and significance of this research. The authors should add more specific references to previous works that directly influenced the choice of coating materials and thicknesses. The research design is well-structured and appropriate but it is well clear the selection criteria for the participants. Moreover, The authors need to provide more details on the rationale behind choosing the specific coating thicknesses for the experiments. They must describe the statistical methods used in more detail, including any assumptions made and the reasons for choosing specific tests. They should include more comparative data to emphasize the differences between the coated and standard vests. In the conclusions should discuss potential limitations of the study and highlight the practical implications for the design and use of cooling vests in different industries.

Author Response

Comment 1: The paper is interesting but the introduction provides a comprehensive background. However, the authors have to Include a brief discussion on the limitations of previous studies on PDRC to better highlight the novelty and significance of this research.

Reply 1: Thank you for your valuable suggestion. The limitations of previous studies on PDRC have been discussed in the Introduction. 

Although PDRC materials have been developed and their cooling properties were tested, few studies have examined the integration of PDRC materials with basic clothing to assess human thermal comfort. Inspired by the above research, this work develops a novel radiative cooling vest with the coating strategy, and the thermal load of the outdoor environment is evaluated with the WBGT index. Through the experiment of radiative cooling performance in a high-temperature outdoor environment, the optimal thickness of the coating is determined to modify the surface property of the vest. The cooling performance of the vest with radiative cooling is experimentally investigated compared with the standard vest. In addition, the TSV and TCV are collected and analyzed to evaluate the human thermal comfort after the outdoor human experiment at different WBGTs.”

Comment 2: The authors should add more specific references to previous works that directly influenced the choice of coating materials and thicknesses.

Reply 2: Thank you for the suggestion. The specific references to previous works that directly influenced the choice of coating materials and thicknesses have been added in the Introduction.

“Two cooling approaches are widely used to maintain thermal comfort. One is active cooling, which achieves an excellent cooling effect at the cost of energy consumption [5]. Specifically, ventilation and air conditioning systems used for active space cooling typically have high capital and operating costs, which are impractical for outdoor services [6]. Liquid cooling [7,8], air cooling [9–11], semiconductor cooling [12,13], phase change material cooling [14] textiles and wearables used for personal active cooling consume less energy. Still, current wearables are bulky and require regular replacement of accessories, making them unsuitable for outdoor workers to wear for a long time. The other is passive daytime radiative cooling (PDRC), which is very promising for personal thermal management because of its excellent cooling effect [15]. It radiates heat to outer space and reflects the sunlight without extra power to achieve cooling [16–22]. At present, various intelligent fabrics enabled by nanofiber membranes, microfiber fabric, and coated fabric have been successfully utilized in personal thermal management. Zhang et al. [23] developed a passive cooling layered metamaterial fabric based on a nanofiber membrane for moisture absorption and perspiration. The metamaterial fabric showed high solar reflectance emissivity and remarkable moisture permeability. The human body temperature covered by metamaterial cooling fabric developed by Zeng et al. [24] was about 4.8 °C lower than that covered by cotton fabric. Cheng et al. [25] prepared a radiative cooling coating with the biomimetic structure of human natural skin wrinkles, achieving a sub-ambient cooling temperature of 8.1 °C at most. Compared to other nano-based methods, nano-coating for surface modification offers strong operability at a low cost while achieving excellent cooling effects. Figure 1 shows the working schematic for radiative cooling coated fabrics. When sunlight strikes the coated fabric, it cools itself by reflecting the sunlight and emitting heat energy to the cold outer space through the atmospheric window. Zhang et al. [26] employed a simple and scalable spraying strategy to fabricate colorful radiative cooling cotton fabrics, which achieved a temperature difference of 5.6 °C compared to bare cotton fabrics. Xiao et al. [27] enhanced the cooling effects of safety helmets with a radiative cooling coating. Under direct sunlight, the internal temperature of the improved helmet is lower than that of the ordinary white helmet, and the maximum temperature difference is 12.2 °C. Zhang et al. [28] experimentally compared the thermal comfort of two reduced-size tent models. The results indicated that the average temperature in the tent model using reflective materials was 4.8 °C lower than that of a regular tent. In addition, the PDRC materials attached to the fabric, and the coating thickness are crucial factors in determining the cooling effect of coated fabrics [19]. Zhong et al. [29] fabricated a multifunctional cotton fabric for radiative outdoor cooling using spectrally selective aluminum phosphate particles. Since the thicker coating results in uneven pore sizes and stiffening of the textile, Cui et al. [30] selected the ideal thickness when considering the preparation process and the flexibility of the coated textile.

References

[19] Dong, Y.; Han, H.; Wang, F.; Zhang, Y.; Cheng, Z.; Shu, X.; Yan, Y. A Low-cost Sustainable Coating: Improving Passive Daytime Radiative Cooling Performance Using the Spectral Band Complementarity Method. Renew. Energ. 2022, 19, 606-616.

[29] Zhong, S.; Yi, L.; Zhang, J.; Xu, T.; Xu L.; Zhang X.; Zuo, T.; Cai, Y. Self-cleaning and Spectrally Selective Coating on Cotton Fabric for Passive Daytime Radiative Cooling. Chem. Eng. J. 2021, 407, 127104.

[30] Cui, C.; Lu, J.; Zhang, S.; Han, J. Hierarchical-porous Coating Coupled with Textile for Passive Daytime Radiative Cooling and Self-cleaning. Sol. Energy Mater Sol. Cells, 2022, 247, 111954.

Comment 3: The research design is well-structured and appropriate but it is well clear the selection criteria for the participants. Moreover, The authors need to provide more details on the rationale behind choosing the specific coating thicknesses for the experiments.

Reply 3: Thank you for your valuable suggestion. We have added the selection criteria for the participants and the rationale behind choosing the specific coating thicknesses for the experiments in the manuscript.

Before the experiments, the minimum number of participants was estimated to be six (total sample size/number of measurements) using the G*Power software. The effect size, α and statistical power (1-β) are 0.4, 0.05, and 0.8 respectively (α and β are the probability of type I and II errors, respectively). The subjects selected for the experiment are all healthy men around 23 years old. Six healthy college students (age: 23±1 years old, height: 176±3 cm, weight: 69±7 kg, BMI: 21.0±2.7 kg/m2) are involved in the subjective questionnaires on thermal comfort at four different WBGTs. The participants are informed not to take any alcohol, caffeine, or medical drugs for at least 24 hours before the experiments. The subjects wear the same basic outfit under a vest, including a cotton T-shirt, knee-high shorts, socks, underwear, and thin sneakers. The clothing insulation is approximately 0.5 clo [34]. The study is approved by the University Ethics Committee and all participants provide written informed consent before the experiments. The guidelines of the Declaration of Helsinki are followed in the experiment [35]. The personal information of the subjects is shown in Table 2.”

“An outdoor experiment is conducted to compare the cooling performance of the fabrics with coatings of different thicknesses. The measured meteorological parameters, the inner surface temperature of the fabric, and the air temperature inside the test box are shown in Figure 7(a). The fabrics with coating thicknesses of 160 µm and 220 µm have lower temperatures than those of the other coated fabrics at most of the time. The fabric temperature is reduced by 12.22 °C maximumly with the coating of 160 µm in thickness, compared with the air temperature inside the test box. The average temperatures of the fabrics with different coating thicknesses are shown in Figure 7(b). It is seen that the coating with a thickness of 160 um has the maximum average temperature, 5.87 °C lower than the air temperature inside the test box. It is concluded that the fabric with a coating thickness of 160 µm possesses the optimal radiative cooling performance. By the time fabrics are exposed to the sun for 170 minutes, the temperatures of the inner surface of fabrics decrease significantly due to the reduction of solar radiation intensity (see the red circle in Figure 7(a)). The decrease in solar radiation intensity is caused by a sudden increase in cloud cover in the sky. In addition, the fabric with a coating thickness of 160µm is softer than the fabric with a coating thickness of 220µm and 300µm, and the coated fabric surface possesses higher smoothness and better uniform continuity. From the above experimental results, it is seen that the fabric with a coating thickness of 160µm shows a higher cooling effect. At the same time, a good economy for preparing the coated fabric is clearly reflected compared with thicker coating fabrics.

 Comment 4: They must describe the statistical methods used in more detail, including any assumptions made and the reasons for choosing specific tests.

Reply 4: Thank you for your valuable suggestion. Thanks for your valuable suggestions. We have described the statistical methods used in more detail as follows:

“SPSS statistics software (version 26) is used in the statistical analysis. In the experiment, a one-way analysis of variance is used for comparing the thermal comfort of subjects wearing standard and refrigerated vests under each specific wet bulb globe temperature. The samples of one-way analysis satisfy normality, independence, and homogeneity of variance. Results are considered statistically significant at p<0.05. To assess the statistical significance of differences in thermal comfort between standard vests and cooling vests, a paired-sample t-test is performed to analyze the significant difference in relevant variables under the standard vest and cooling vest conditions. The samples of the paired t-test satisfy the pairing condition, have normally distributed differences, exhibit independence between pairs, and assume equal variance of the differences. The symbol "*" indicates a significant difference (p<0.05), and the symbol "**" indicates a highly significant difference (p<0.01) in the figures.”

Comment 5: They should include more comparative data to emphasize the differences between the coated and standard vests.

Reply 5: Thanks for your valuable suggestions. We have provided more comparative data about the differences between the coated and standard vests as follows: 

Table 1. The comparison for the coated and standard vests

Type Coating thickness (um) Atmospheric window emissivity (%) Reflectivity (%) Absorptivity (%)
Coated vests 160 90.2 82.5 17.5
Standard vests 0 89.7 65.1 34.9

Comment 6: In the conclusions should discuss potential limitations of the study and highlight the practical implications for the design and use of cooling vests in different industries.

Reply 6: Thank you for your valuable suggestion. We have added discussion about the potential limitations of the study and highlighted the practical implications in the conclusions as follows:

“This study provided a radiative cooling vest based on a working suit, which was easily made at a low cost. The human thermal comfort was evaluated in the condition of wearing radiative cooling vests and standard vests. The radiative cooling performance was tested for the coated fabrics with different thicknesses. Radiative cooling vests and standard vests were compared, concerning the inner and outer surface temperatures in an outdoor high-temperature environment. The radiative cooling vest significantly improved human thermal comfort. The present study leads to the following conclusions:

(1) The fabric with a thickness of 160 µm has a remarkable radiative cooling effect with an average temperature of 5.87 °C lower than the ambient temperature.

(2) Compared with the standard vest, the maximum temperatures at the inner and outer surfaces of the radiative cooling vest are reduced by 5.54 °C and 4.37 °C, respectively.

(3) The thermal comfort of the human body is improved by wearing the radiative cooling vests at different wet bulb globe temperatures, and the degree of improved cooling decreases with the increase of wet bulb globe temperature.

(4) The significant improvement of human thermal comfort is achieved at a wet bulb globe temperature of 26 °C. Compared with the standard vest, the percentage of thermal sensation vote in the range of 0~1 increases from 29.2% to 66.7%, and the mean thermal comfort vote increases from about -0.5 to 0.2 by applying the radiative cooling coating to the vest.

The cooling vest developed in this work still suffers from poor air permeability, and the cooling effect requires further enhancement. To address these issues, it may be necessary to develop multiple types of cooling clothing, such as adding fans or developing new cooling materials. However, the significant improvement in comfort provided by the cooling vest is sufficient for its application to departments whose work activities are carried out outdoors, such as construction workers, traffic police, cleaners, and other outdoor workers. The purpose of removing heat from the human body and improving human performance is achieved by wearing a cooling vest which represents wearable technology.

Round 2

Reviewer 2 Report

Comments and Suggestions for Authors

Accept